# Robust Preference Optimization through Reward Model Distillation

## Abstract

Language model (LM) post-training (or alignment) involves maximizing a reward function that is derived from preference annotations. Direct Preference Optimization (DPO) is a popular offline alignment method that trains a policy directly on preference data without the need to train a reward model or apply reinforcement learning. However, typical preference datasets have only a single, or at most a few, annotation per preference pair, which causes DPO to overconfidently assign rewards that trend towards infinite magnitude. This frequently leads to degenerate policies, sometimes causing even the probabilities of the *preferred* generations to go to zero. In this work, we analyze this phenomenon and propose *distillation* to get a better proxy for the true preference distribution over generation pairs: we train the LM to produce probabilities that match the distribution induced by a reward model trained on the preference data. Moreover, to account for uncertainty in the reward model we are distilling from, we optimize against a *family of reward models* that, as a whole, is likely to include at least one reasonable proxy for the preference distribution. Our results show that distilling from such a family of reward models leads to improved robustness to distribution shift in preference annotations, while preserving the simple supervised nature of DPO.

## 1 Introduction

Language model (LM) post-training (or alignment) aims to steer language model policies towards responses that agree with human preferences. Early state-of-the-art approaches have focused on reward learning from human feedback. In this paradigm, preference annotations are used to train reward models, which then guide the optimization of the language model policy through online reinforcement learning (an approach broadly referred to as RLHF). Recent research on offline "Direct Preference Optimization" [DPO; 23] and extensions thereof [3; 31], however, has demonstrated that it is also possible to directly optimize policies on the preference data, which bypasses the need for a separate reward model—and its offline nature also leads to faster, and simpler, training frameworks.

While this direct approach to preference optimization is attractive in terms of its simplicity and efficiency, it also raises important questions about the effectiveness and robustness of the resulting policies—as well as the broader utility of using an explicit reward model. In this paper, we argue that explicit reward modeling can, in fact, offer substantial practical advantages that are not captured by DPO's formulation. In particular, we theoretically show that relying solely on the preference data can be a precarious strategy, with few natural brakes in place to prevent policies trained under the DPO objective from careening off towards degenerate policies when the preference data exhibits certain idiosyncratic properties. On the other hand, explicit reward models can easily be regularized and understood—regardless of whether they are Bradley-Terry models [4], margin-based ranking models [40], or simply any other kind of function that correlates well with human preferences [31; 17].

Taking a step back from pure direct preference optimization, we propose a method that merges the best of both worlds: an efficient reward model distillation algorithm that (i) operates effectively in the offline setting, (ii) makes minimal assumptions about the true, optimal reward we aim to maximize, and (iii) demonstrates greater robustness to the specific distribution of prompt/response data used for policy alignment. Drawing inspiration from prior knowledge distillation techniques [14; 26; 35; 10], we leverage the same change of variables trick employed in DPO to express the language model policy in terms of its implicit reward model [23]. We then train the policy to match our desired, explicit reward via an $L_2$ loss that directly regresses the pairwise differences in target rewards for any two generation pairs $(x, y_1)$ and $(x, y_2)$. We theoretically establish the equivalence between optimizing this distillation loss over a sufficiently diverse offline dataset of unlabeled examples and optimizing the traditional online RLHF objective.

Our reward model distillation approach, however, is not immune to some of the same challenges facing DPO-style learning of policies. In particular, reward model distillation requires having a reliable reward model—but having a reliable reward requires having a reliable method for extracting a reward model from a potentially noisy preference dataset. To address the uncertainty surrounding the "right" reward model, we introduce a pessimistic extension to our approach. This extension aims to maximize the worst-case improvement of our model across a plausible family of reward models (e.g., those sufficiently consistent with annotated preference data). This strategy aligns with that of existing work in conservative offline reinforcement learning [5; 16]. Interestingly, we derive that this pessimistic objective can be equivalently expressed and optimized by adding a simple additional KL-divergence regularization to the original distillation objective.

Empirically, we find that reward model distillation, particularly pessimistic reward model distillation, leads to similar performance to prior direct preference optimization methods in settings where the preference datasets used are unbiased, but significantly better performance in settings where the preference datasets are biased, when compared to DPO and the Identity Preference Optimization (IPO) framework of [3], which was introduced as a more robust alternative to DPO. To further support these empirical observations, we provide an extensive theoretical analysis that both (i) sheds more light on the degenerative tendencies of DPO and issues inherent to its objective, and (ii) highlights relative advantages of our explicitly regularized approaches.

## 2 Preliminaries

We begin with a brief review of Direct Preference Optimization (DPO) [23] and its analysis. Proofs of all theoretical results provided here, and in the rest of the paper, are deferred to Appendix A.

### 2.1 The preference alignment problem

Let $x$ be an input prompt, and let $y \sim \pi_\theta(\cdot \mid x)$ be the language model policy $\pi_\theta$'s response to $x$. Given some reward function $r^*(x, y)$ and another reference policy $\pi_{\text{ref}}(y \mid x)$, the goal of alignment is to solve for the "aligned" policy $\pi_{\theta^*}(y \mid x)$ that maximizes the following RLHF objective, i.e.,

$$\pi_{\theta^*}(y \mid x) = \arg\max_{\pi_\theta} \mathbb{E}_{\mu(x)} \left[ \mathbb{E}_{\pi_\theta(y|x)}[r^*(x, y)] - \beta \mathbb{D}_{\text{KL}}[\pi_\theta(\cdot \mid x) \| \pi_{\text{ref}}(\cdot \mid x)] \right], \quad (1)$$

where $\mu(x)$ is a fixed distribution over prompts, and the KL-divergence term prevents the aligned policy from being dramatically different from the anchoring reference policy, $\pi_{\text{ref}}(y \mid x)$. Here, the reward function $r^*$ is typically not known in advance, but rather inferred from collected human preference data in the form of $(x, y^w, y^\ell)$, where $x$ is the prompt, $y^w$ is the "winning", or preferred, response, and $y^\ell$ is the "losing", or dispreferred, response. A common approach is to assume that pairs $(y_1, y_2)$ follow a Bradley-Terry model [4], under which the probability that $y_1$ is preferred to $y_2$ given the reward function $r^*$ and prompt $x$ is $p^*(y_1 \succ y_2 \mid x) = \sigma(r^*(x, y_1) - r^*(x, y_2))$, where $\sigma(\cdot)$ is the sigmoid function and $\succ$ denotes preference. Under this model, we can use the preference data $(x, y^w, y^\ell) \sim \mathcal{D}_{\text{pref}}$ to estimate $r^*$ via maximum likelihood estimation, i.e.,

$$\hat{r} \in \arg\min_r \mathbb{E}_{(y^w, y^\ell, x) \sim \mathcal{D}_{\text{pref}}} \left[ -\log \sigma(r(x, y^w) - r_\phi(x, y^\ell)) \right]. \quad (2)$$

With $\hat{r}$ in hand, Eq. (1) can be optimized using standard reinforcement learning algorithms [27; 29; 6].

## 2.2 Direct preference optimization

DPO is a simple approach for offline policy optimization that uses preferences to directly align the language model policy, without training an intermediate reward model. Specifically, DPO leverages the fact that the optimal solution to the KL-constrained objective in (1) takes the form [15]

$$\pi_{\theta^*}(y \mid x) = \frac{1}{Z(x)} \pi_{\text{ref}}(y \mid x) \exp\left(\frac{1}{\beta} r^*(x, y)\right), \tag{3}$$

where $Z(x) = \sum_y \pi_{\text{ref}}(y \mid x) \exp(\frac{1}{\beta} r^*(x, y))$ is the partition function. DPO reparameterizes the true reward function $r^*$ in terms of the optimal policy $\pi_{\theta^*}$ that it induces, i.e.,

$$r^*(x, y) = \beta \log\left(\frac{\pi_{\theta^*}(y \mid x)}{\pi_{\text{ref}}(y \mid x)}\right) + \beta \log Z(x). \tag{4}$$

Under the Bradley-Terry model, the likelihood that $y_1 \succ y_2$ can then be written as

$$p^*(y_1 \succ y_2 \mid x) = \sigma\left(\beta \log \frac{\pi_{\theta^*}(y_1)\pi_{\text{ref}}(y_2)}{\pi_{\theta^*}(y_2)\pi_{\text{ref}}(y_1)}\right), \tag{5}$$

where now $\pi_{\theta^*}$ can be directly estimated on $\mathcal{D}_{\text{pref}}$ following the objective in (2), in place of the intermediate reward model $\hat{r}$, i.e., $\pi_{\hat{\theta}}(y \mid x) \in \operatorname{argmin}_{\pi_\theta} \mathcal{L}_{\text{dpo}}(\pi_\theta; \mathcal{D}_{\text{pref}})$ where

$$\mathcal{L}_{\text{dpo}}(\pi_\theta; \mathcal{D}_{\text{pref}}) = \mathbb{E}_{(y^w, y^\ell, x) \sim \mathcal{D}_{\text{pref}}}\left[-\log \sigma\left(\beta \log \frac{\pi_{\theta^*}(y^w)\pi_{\text{ref}}(y^\ell)}{\pi_{\theta^*}(y^\ell)\pi_{\text{ref}}(y^w)}\right)\right]. \tag{6}$$

## 2.3 Pitfalls of direct preference optimization

As argued in [3], the Bradley-Terry assumption that DPO strongly relies on for maximum likelihood estimation is sensitive to the underlying preference data. Specifically, if we have any two responses $y_1$ and $y_2$ where $p^*(y_1 \succ y_2 \mid x) = 1$, then the Bradley-Terry model dictates that $r^*(y_1) - r^*(y_2) = +\infty$, and therefore $\pi_{\theta^*}(y_2 \mid x) = 0$ for *any* finite KL-regularization strength $\beta$.

We can illustrate this phenomenon on a broader level with the following example.

**Assumption 1.** *Suppose we are given a preference dataset of (context-free) pairs $\mathcal{D}_{\text{pref}} = \{(y_i^w, y_i^\ell)\}_{i=1}^n$, the pairs $(y_i^w, y_i^\ell)$ are mutually disjoint in both the elements. Further suppose that we optimize the DPO objective on $\mathcal{D}_{\text{pref}}$ with a single parameter $\theta_y$ for each $y$.*

**Proposition 1.** *Under Assumption 1, for any $(y, y')$ such that $y = y_i^w$ and $y' = y_i^\ell$ for some $i$, we have $\frac{\pi_{\theta^*}(y)\pi_{\text{ref}}(y')}{\pi_{\theta^*}(y')\pi_{\text{ref}}(y)} \to \infty$, for all global minimizers $\pi_{\theta^*}$ of the DPO objective in (6), for any $\beta > 0$.*

**Corollary 1.** *Under Assumption 1, further assume that $0 < \pi_{\text{ref}}(y) < 1$ for all $y$. Then $\pi_{\theta^*}$ is a global minimizer of the DPO objective in (6) iff $\pi_{\theta^*}(\mathcal{C}(y^\ell)^c) \to 1$ with $\pi_{\theta^*}(y_i^w) > 0 \, \forall i \in [n]$, where $\mathcal{C}(y^\ell)^c$ is the complement of the set of all responses $y$ that appear as a dispreferred $y_i^\ell$ for any $i \in [n]$.*

Additional analysis of the training dynamics of DPO is also provided in §5. A significant, and non-obvious, implication of Corollary 1 is that the set of global optima of the DPO loss also includes policies that can shift nearly all probability mass to responses that never even appear in the training set—and even assign near zero probability to all of the training data responses that do in fact correspond to winning generations, $y^w$, a phenomenon that has been observed empirically [e.g., 20]. Stated differently, Corollary 1 implies that any $\theta^*$ merely satisfying $\pi_{\theta^*}(y_i^\ell) = 0$ with $\pi_{\theta^*}(y_i^w) > 0 \, \forall i \in [n]$ is a global minimizer of the DPO objective in this setting. Though simplistic, the scenario in Assumption 1 is closer to reality than might first be appreciated: in many practical situations we can almost always expect the finite-sample preference data to contain one (or at most a few) preference annotations per example $(x, y_1, y_2)$, while the policies $\pi_\theta$ can have billions of parameters ($\gg n$). Of course, this issue can also be viewed as a classic instance of overfitting—with the additional caveat that as opposed to *overpredicting* responses within the training set, we might overfit to *almost never* producing anything like the "good" responses that do appear within the training set. Furthermore, without additional regularization (beyond $\beta$), we can expect this degeneration to easily happen in typical preference datasets.

## 3 Uncertainty-aware reward model distillation

As discussed in the previous section, a core issue in preference optimization is that the true preference distribution $p^*(y_1 \succ y_2 \mid x)$ is not known. Attempting to infer it from finite-sample preference data (that may further be biased or out-of-distribution with respect to the target domain) can then result in a failure to learn reasonable policies. In this section, we now propose an inherently regularized approach to direct preference optimization that uses uncertainty-aware reward model distillation.

### 3.1 Reward model distillation

Suppose for the moment that the reward function $r^*$ was in fact known, and did not have to be inferred from sampled preference data. Under this setting, we can then define an efficient offline optimization procedure that is similar in spirit to DPO, but no longer relies directly on a preference dataset. Concretely, given unlabeled samples $(x, y_1, y_2) \sim \rho$ (where the number of samples can be potentially unlimited), we can define a simple "distillation" loss, $\mathcal{L}_{\mathrm{distill}}(r^*, \pi_\theta)$, as follows:

$$\mathcal{L}_{\mathrm{distill}}(r^*, \pi_\theta; \rho) = \mathbb{E}_{\rho(x, y_1, y_2)} \left[ \left( r^*(x, y_1) - r^*(x, y_2) - \beta \log \frac{\pi_\theta(y_1 \mid x)\pi_{\mathrm{ref}}(y_2 \mid x)}{\pi_\theta(y_2 \mid x)\pi_{\mathrm{ref}}(y_1 \mid x)} \right)^2 \right]. \quad (7)$$

Intuitively, the distillation loss seeks to exactly match *differences* in reward model scores across all generation pairs $(x, y_1, y_2)$. It is then easy to see that under the Bradley-Terry model, this is equivalent to matching the strength of the preference relationship, $y_1 \succ y_2$. Furthermore, by only matching differences, we can still conveniently ignore the log partition term, $\log Z(x)$, in the implicit reward formulation for $\pi_\theta$ as shown in (4), as it is constant across different $y$ for any given $x$. Finally, similar to the motivation in DPO, we can show that minimizing $\mathcal{L}_{\mathrm{distill}}(r^*, \pi_\theta; \rho)$ indeed results in an optimally aligned policy $\pi_{\theta^*}$, as long as the data distribution $\rho$ has sufficient support.

**Theorem 1.** *Let $\mathcal{Y}$ denote the set of all possible responses for any model $\pi_\theta$. Assume that $\mathrm{supp}(\pi_{\mathrm{ref}}(y \mid x)) = \mathcal{Y}$, i.e., the reference policy may generate any outcome with non-zero probability. Further, let $\mathrm{supp}(\rho(x, y_1, y_2)) = \mathrm{supp}(\mu(x)) \times \mathcal{Y} \times \mathcal{Y}$. Let $\pi_{\theta^*}(y \mid x) \in \mathrm{argmin}_{\pi_\theta} \mathcal{L}_{\mathrm{distill}}(r^*, \pi_\theta; \rho)$ be a minimizer over all possible policies, of the implicit reward distillation loss in (7), for which $r^*(x, y)$ is assumed to be deterministic, and finite everywhere. Then for any $\beta > 0$, $\pi_{\theta^*}$ also maximizes the alignment objective in (1).*

The above result holds for a broad class of data distributions $\rho(x, y_1, y_2)$, and makes no assumptions on $r^*$ (e.g., it is no longer necessary for it to be defined using a Bradley-Terry model). In fact, this result can also be seen as strict generalization of the IPO framework of [3] when taking $r^*(x, y) \triangleq \mathbf{1}\{y = y_w\}$, if labeled pairs $(x, y_w, y_l)$ are provided instead of the unlabeled pairs $(x, y_1, y_2)$.

Of course, the true reward $r^*$ is usually not known in practice. Still, as in standard RLHF, we can go about constructing good proxies by using the preference data to identify plausible target reward models $r_{\mathrm{tgt}}$—further guided by any amount of regularization and inductive bias that we desire. A natural choice is to first learn $r_{\mathrm{tgt}}$ on the preference data $\mathcal{D}_{\mathrm{pref}}$ using standard methods, and then reuse $\mathcal{D}_{\mathrm{pref}}$ to distill $\pi_\theta$, which is similar to classical settings in teacher-based model distillation [14; 26]. Furthermore, as $r_{\mathrm{tgt}}$ is a real-valued model, at a bare minimum it is guaranteed to induce a regularized Bradley-Terry preference distribution $p_{\mathrm{tgt}}(y_1 \succ y_2 \mid x) > 0$, $\forall x, y_1, y_2 \in \mathcal{X} \times \mathcal{Y}$, and thereby avoid some of the degeneracies identified in §2.3 for the maximum likelihood estimate under DPO.

### 3.2 Pessimistic reward model distillation

Choosing a single reward model $r_{\mathrm{tgt}}$ for anchoring the LM policy can naturally still lead to degenerate behavior if $r_{\mathrm{tgt}}$ is a poor approximation of the true $r^*$ that accurately reflects human preferences. However, we can easily extend our framework to handle uncertainty in the right target reward function by defining a confidence *set* of $k \geq 1$ plausible target reward models, $\mathcal{S} = \left\{ r_{\mathrm{tgt}}^1, \ldots, r_{\mathrm{tgt}}^k \right\}$, and training $\pi_{\theta^*}(y \mid x)$ to maximize the following "pessimistic" form of the objective in (1):

$$\max_{\pi_\theta} \min_{r_{\mathrm{tgt}}^i \in \mathcal{S}} \mathbb{E}_{\mu(x)} \left[ \underbrace{\mathbb{E}_{\pi_\theta(y|x)}[r_{\mathrm{tgt}}^i(x, y)] - \mathbb{E}_{\pi_{\mathrm{ref}}(y|x)}[r_{\mathrm{tgt}}^i(x, y)]}_{\text{advantage over baseline policy}} - \beta \mathbb{D}_{\mathrm{KL}}(\pi_\theta(\cdot \mid x) \| \pi_{\mathrm{ref}}(\cdot \mid x)) \right]. \quad (8)$$

In this pessimistic objective we are no longer op-
timizing $\pi_\theta$ for a single reward, but optimizing
$\pi_\theta$ to produce generations that are scored favor-
ably on average, even by the worst-case reward
model in the set $\mathcal{S}$, relative to the generations of
the baseline policy $\pi_{\text{ref}}$. When the set $\mathcal{S} = \{r^*\}$
consists of only the ground-truth reward, the ob-
jective (8) is equivalent to standard RLHF (1),
up to a constant offset independent of $\theta$. More
generally, whenever $\mathcal{S}$ includes a good proxy
$\tilde{r}$ for $r^*$, the pessimistic advantage evaluation
ensures that the the policy $\pi_\theta^*$ that maximizes
eq. (8) still has a large advantage over $\pi_{\text{ref}}$ under
all $r \in \mathcal{S}$, including $\tilde{r}$. This use of pessimism
to handle uncertainty in the knowledge of the
true reward is related to similar techniques in
the offline RL literature [16; 5].

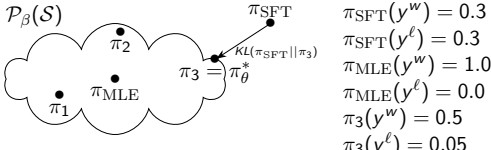

Figure 1: A toy illustration of Theorem 2, which
states that the optimal $\pi_{\theta^*}$ for (8) is the policy
in $\mathcal{P}_\beta(\mathcal{S})$ with the lowest forward-KL from $\pi_{\text{SFT}}$.
The set $\mathcal{P}_\beta(\mathcal{S})$ contains a (potentially infinite) set
of policies $\pi_1, \pi_2, \ldots$ corresponding to target re-
ward models. Here, $\pi_{\text{SFT}}$ assigns equal mass to $y^w$
and $y^\ell$, $\pi_{\text{MLE}}$ is the MLE solution for the DPO ob-
jective, which puts all probability mass on $y^w$, and
$\pi_3$ is the policy in $\mathcal{P}_\beta(\mathcal{S})$ with lowest forward-KL.

For the objective to be meaningful, the set $\mathcal{S}$ has to be chosen carefully. When $\mathcal{S}$ is small, it might
not include any good proxy for $r^*$. Conversely, if $\mathcal{S}$ is too rich, it forces $\pi_{\theta^*}$ to be nearly identical to
$\pi_{\text{ref}}$, since any deviations from $\pi_{\text{ref}}$ might be penalized by some reward model in $\mathcal{S}$. Consequently,
we want to design $\mathcal{S}$ to be the smallest possible set which contains a reasonable approximation to $r^*$.

To optimize (8), it turns out that we can formulate it as an equivalent constrained offline optimization
problem, that we will show to conveniently admit a similar loss form as (7).

**Theorem 2** (Pessimistic distillation). *Define the constrained minimizer*

$$\pi_{\theta^*}(y \mid x) \in \underset{\pi_\theta \in \mathcal{P}_\beta(\mathcal{S})}{\operatorname{argmin}} \; \beta \mathbb{E}_{\mu(x)} \mathbb{D}_{\text{KL}}(\pi_{\text{ref}}(\cdot \mid x) \| \pi_\theta(\cdot \mid x)), \tag{9}$$

*where $\mathcal{P}_\beta(\mathcal{S})$ is the set of all possible policies with implicit reward models that are consistent with
any target reward model $r_{\text{tgt}}^i \in \mathcal{S}$, i.e., $\mathcal{P}_\beta(\mathcal{S}) \triangleq \{\pi_{\theta_i}\}_{i=1}^{|\mathcal{S}|}$ where $\pi_{\theta_i} \propto \pi_{\text{ref}}(y \mid x) \exp \frac{1}{\beta} r_{\text{tgt}}^i(x, y)$.
Then for any $\beta > 0$, $\pi_{\theta^*}$ also maximizes the pessimistic alignment objective in (8).*

To unpack this result, Theorem 2 stipulates that the $\pi_\theta$ that maximizes the pessimistic objective in (8)
is the policy in $\mathcal{P}_\beta(\mathcal{S})$ that is closest in *forward* KL-divergence to $\pi_{\text{ref}}$ (see Figure 1).[1] In addition,
this policy also maximizes the expected reward of one of the $r_{\text{tgt}}^i \in \mathcal{S}$ (minus the additional weighted
reverse KL-divergence penalty term). Intuitively, the forward KL-divergence term serves the role of
biasing the model towards optimizing for reward models that are similar to the implicit reward that
$\pi_{\text{ref}}$ already maximizes. Otherwise, there might exist a target reward model $r_{\text{tgt}}^i \in \mathcal{S}$ for which the
advantage of $\pi_\theta$ relative to $\pi_{\text{ref}}$ will be low, or even negative (a solution that we would like to avoid).

### 3.2.1 Optimization

The constraint in (9) can then be relaxed and approximately optimized by introducing an objective
with a Lagrangian-style penalty with strength $\alpha > 0$ on a form of distillation loss as (7), i.e.,

$$\min_{\pi_\theta} \beta \mathbb{E}_{\mu(x)} \mathbb{D}_{\text{KL}}(\pi_{\text{ref}}(y \mid x) \| \pi_\theta(y \mid x)) + \alpha \min_{r_{\text{tgt}}^i \in \mathcal{S}} \mathcal{L}_{\text{distill}}(r_{\text{tgt}}^i, \pi_\theta; \rho), \tag{10}$$

where in practice we divide by $\alpha$ and instead optimize[2]

$$\mathcal{L}_{\text{pdistill}}(\mathcal{S}, \pi_\theta; \rho) = \min_{r_{\text{tgt}}^i \in \mathcal{S}} \mathcal{L}_{\text{distill}}(r_{\text{tgt}}^i, \pi_\theta; \rho) + \gamma \mathbb{E}_{\mu(x)} \mathbb{D}_{\text{KL}}(\pi_{\text{ref}}(\cdot \mid x) \| \pi_\theta(\cdot \mid x)), \tag{11}$$

where $\gamma = \beta \alpha^{-1}$. In reality, minimizing (11) for $\gamma > 0$ is equivalent to solving the constrained
optimization problem in (9) with an implicitly larger set of possible reward models $\mathcal{S}_\gamma \supseteq \mathcal{S}$ indexed
by $\gamma$. More specifically, $\mathcal{S}_\gamma$ also contains all reward models $\tilde{r}$ that are approximately consistent with
the anchoring reward models $r_{\text{tgt}}^i$ contained in $\mathcal{S}$, as the following result states.

---

[1]Note that the objective in (9) minimizes the *forward* KL-divergence $\mathbb{D}_{\text{KL}}(\pi_{\text{ref}}(\cdot \mid x) \| \pi_\theta(\cdot \mid x))$ even
though the pessimistic objective in (8) is regularized with *reverse* KL-divergence $\mathbb{D}_{\text{KL}}(\pi_\theta(\cdot \mid x) \| \pi_{\text{ref}}(\cdot \mid x))$.

[2]In practice, we compute and optimize the min over reward models per each mini-batch of examples.

**Proposition 2** (Soft pessimistic distillation). *Assume the same conditions as Theorem 1. Then for any $0 < \gamma < \infty$, there exists a $\lambda \geq 0$ such that $\pi_{\theta^*}(y \mid x) \in \arg\min_{\pi_\theta} \mathcal{L}_{\mathrm{pdistill}}(\mathcal{S}, \pi_\theta; \rho)$, where $\pi_{\theta^*}$ is a minimizer over all possible policies, is a solution to (9) for the effective reward model set*

$$\mathcal{S}_\gamma = \bigcup_{r^i_{\mathrm{tgt}} \in \mathcal{S}} \left\{ \tilde{r} \colon \mathbb{E}_{\rho(x,y_1,y_2)} \left[ (r^i_{\mathrm{tgt}}(x,y_1) - r^i_{\mathrm{tgt}}(x,y_2) - \tilde{r}(x,y_1) + \tilde{r}(x,y_2))^2 \right] \leq \lambda \right\}. \quad (12)$$

As a result, optimizing (11) even when using the singleton $\mathcal{S} = \{r_{\mathrm{tgt}}\}$ yields an implicitly pessimistic objective, in which the pessimism is over all reward models $\tilde{r}$ that are consistent up to $\lambda$ with $r_{\mathrm{tgt}}$.

### 3.3 Pessimistic DPO

We can also observe that Proposition 2 can be leveraged to obtain an alternative, implicitly pessimistic, objective that uses DPO directly instead of distillation. Consider the following regularized DPO loss:

$$\mathcal{L}_{\mathrm{pdpo}}(\pi_\theta; \mathcal{D}_{\mathrm{pref}}) = \mathcal{L}_{\mathrm{dpo}}(\pi_\theta; \mathcal{D}_{\mathrm{pref}}) + \gamma \mathbb{E}_{\mu(x)} \mathbb{D}_{\mathrm{KL}}(\pi_{\mathrm{ref}}(y \mid x) \| \pi_\theta(y \mid x)). \quad (13)$$

Following a similar analysis as in Proposition 2, we can derive that this implicitly corresponds to maximizing the pessimistic objective in (8) for the reward model set

$$\mathcal{S}_\gamma = \left\{ r_{\pi_\theta} \colon \mathcal{L}_{\mathrm{dpo}}(\pi_\theta; \mathcal{D}_{\mathrm{pref}}) \leq \min_{\pi'_\theta} \mathcal{L}_{\mathrm{dpo}}(\pi'_\theta; \mathcal{D}_{\mathrm{pref}}) + \lambda \right\}, \quad (14)$$

where $r_{\pi_\theta}(x,y) \triangleq \beta \log \pi_\theta(y \mid x)/\pi_{\mathrm{ref}}(y \mid x) + \beta \log Z(x)$ is the implicit reward model defined by $\pi_\theta$. $\mathcal{S}_\gamma$ then corresponds to the set of reward models $r_{\pi_\theta}$ that are all approximate minimizers of the DPO loss. This not only includes the MLE, but also all other estimators that obtain nearly the same loss. In principle, this can be expected to help ameliorate some of the issues of §2.3: since driving the reward to $\pm\infty$ only marginally decreases the $\mathcal{L}_{\mathrm{dpo}}$ loss past a certain point, the set $\mathcal{S}$ will also include finite reward functions $|r_{\pi_\theta}(x,y)| < \infty$ for any $\gamma > 0$. These rewards would then be preferred if they induce a policy with a smaller (forward) KL-divergence to $\pi_{\mathrm{ref}}$ than the degenerate, infinite rewards.

## 4 Experimental results

The main motivation for reward distillation and pessimism is to increase alignment robustness in challenging settings where it is difficult to learn good policies directly from the preference data. To demonstrate the effectiveness of our approach, we run experiments on the popular TL;DR summarization task [29; 32], in which we simulate a scenario where the preference data has a spurious correlation between the *length* of a summary and whether or not it is preferred.[3]

### 4.1 Experimental setup

We first train an "oracle" reward model on the TL;DR preference data training set [29] and relabel all preference pairs with this oracle. This enables us to use the oracle reward model for evaluation, without worrying about the gap to true human preferences. After relabeling, longer responses (where longer is defined as $y_1$ having at least 10% more tokens than $y_2$) are preferred in 61% of the examples.

To test the effect of a spurious correlation on preference-based policy optimization, we select as a training set 30K examples from the relabeled data such that the longer output is preferred in $\rho$ fraction of examples, with $\rho \in \{0.2, 0.3, 0.4, 0.5, 0.6, 0.7, 0.8\}$. Each such training set is denoted $\mathcal{D}_\rho$. At each $\mathcal{D}_\rho$, we compare our approach to DPO [23] and IPO [3], which are currently the most commonly used offline alignment methods. We test the following variants of distillation and pessimism:

- **Distilled DPO** (d-DPO): Trains a reward model $r_\rho$ on $\mathcal{D}_\rho$, and then optimizes $\mathcal{L}_{\mathrm{distill}}(r_\rho, \pi_\theta; \rho)$.

- **Pessimistic DPO** (p-DPO): A pessimistic version of DPO as described in §3.3, trained on $\mathcal{D}_\rho$.

- **Pessimistic Distilled DPO** (pd-DPO): Combines the above two by training a reward model $r_\rho$ on $\mathcal{D}_\rho$ and optimizing the pessimistic distillation objective (Eq. (11)) with confidence set $\mathcal{S} = \{r_{\mathrm{tgt}}\}$.

- **Pessimistic Ensemble DPO** (e-DPO): To create ensembles of reward models, we subsample from each $\mathcal{D}_\rho$ five preference datasets, $\mathcal{D}_{\rho,b}$, at $b \in \mathcal{B} = \{0.2, 0.4, 0.5, 0.6, 0.8\}$, such that the fraction

---

[3]Length has been repeatedly shown in the past to correlate with reward [28; 21].

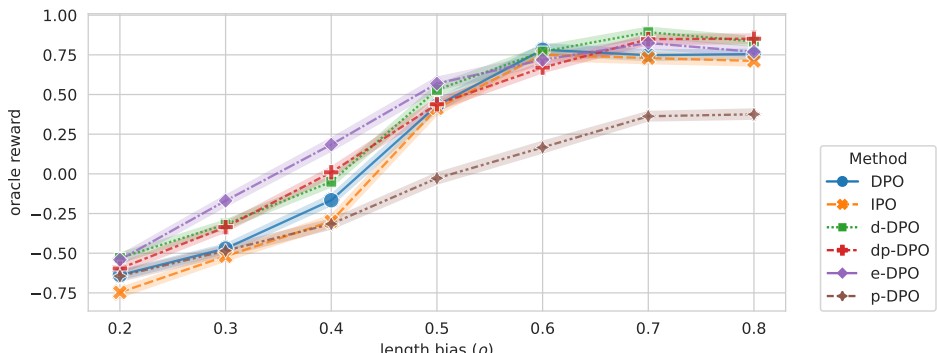

Figure 2: **Main results**, showing the advantage in oracle reward compared to the initial finetuned policy. Errorbars correspond to bootstrap 95% confidence intervals for finite sample variance. Ensemble DPO (e-DPO) is significantly better than DPO and IPO in the challenging setup where shorter responses are preferred ($\rho \leq 0.5$), and is generally the best-performing method overall in this regime. Distilled DPO (d-DPO) performs best when longer responses are preferred ($\rho > 0.6$).

of pairs where the longer response is preferred is $b$, and train reward models $r_{\rho,b}$ on those subsets. Consequently, sensitivity to length should vary across ensemble members. We then apply the same procedure as pd-DPO above, with a confidence set $\mathcal{S}_\rho = \{r_{\rho,b}\}_{b=1}^{\mathcal{B}}$.

All reward models and policies are initialized from Palm-2-XS [2]. Policies also go through a supervised finetuning step on human-written summaries from the original TL;DR training set [32] prior to alignment, and we term this policy $\pi_{\text{SFT}}$. We evaluate performance by sampling summaries for test set prompts, evaluating the average reward according to the oracle reward model, and computing the advantage in average reward compared to $\pi_{\text{SFT}}$ (before alignment). We train policies for $10^4$ steps with batch size 16 and learning rate $10^{-6}$, and reward models for $3k$ steps with batch size 64 and learning rate $4 \times 10^{-6}$. We use the validation set for model selection during policy training and to choose the following hyperparameters. For all DPO variants, we sweep over $\beta \in \{.01, .1, 1, 3, 10, 30, 100\}$. For IPO, we sweep over $\tau \in \{0.01, 0.1, 1, 3, 5, 10, 25\}$. For all pessimistic methods we anneal $\gamma = \alpha/\beta$ from $10^{-4}$ to $10^{-2}$ linearly during the $10k$ training steps.

## 4.2 Results

We present the results of our experiment in Figure 2. As can be seen in the plot, the more challenging setting is when $\rho < 0.5$, which corresponds to a sample of preference annotations in which shorter outputs are generally preferred. This distribution shift is more difficult because as mentioned the oracle reward model (trained on human annotations) has a bias in favor of longer outputs [28]. Nevertheless we get sizable improvements compared to the reference policy $\pi_{\text{SFT}}$ for all length bias values.

All approaches that invoke distillation (d-DPO, e-DPO, dp-DPO) outperform IPO and DPO ($p < .01$ by a Wald test) for $\rho \leq 0.5$, where shorter responses are preferred. Pessimistic ensemble DPO (e-DPO) performs particularly well in these settings, generally outperforming all methods that use a single reward model. When longer responses are preferred ($\rho > 0.6$), single reward distillation (d-DPO) leads to the highest performance, significantly outperforming both DPO and IPO ($p < .01$ by a Wald test). Interestingly, p-DPO does not provide empirical benefits relative to the distillation based methods, indicating that the distillation loss itself is quite important. For the effect of hyper-parameter selection, see Figure D.1. In DPO-based methods, the optimal value of $\beta$ is inversely correlated with the bias; in IPO the same holds for the $\tau$ hyperparameter.

To better understand the utility of reward ensembles in e-DPO, in particular when $\rho < 0.5$, we examine the role of each reward model in the ensemble across different biases. Specifically, given the final e-DPO policy per length bias, for each example we identify the reward model $r_{\rho,b}$ that best matches the implicit reward of this policy, i.e., for which reward model is $\mathcal{L}_{\text{distill}}$ minimized on that example (see Eq. (7) and (11)). We find that when the policy is trained on data where shorter preference are preferred ($\rho < .5$), the reward model that best matches the policy often has the opposite bias ($b$ is high), and vice versa. Thus, the success of e-DPO may be explained by its ability to distill from reward models that do not suffer from the bias in the policy training data, which is particularly

helpful when $\rho \leq .5$ as this bias is also not shared by the oracle RM. We provide the full distribution over reward models for all $\rho$ and $\beta$ in App. C. Overall, these results demonstrate the efficacy of training a policy by distilling from a reward model in the presence of distribution shifts, and that a careful design of an ensemble to mitigate spurious correlations can lead to further performance gains.[4]

## 5    Theoretical analysis

This section characterizes problems with the DPO objective and solutions offered by pessimistic DPO and distillation, focusing on the simplified scenario in which we optimize with respect to a single preference pairs $(y^w, y^\ell)$. Once again, all proofs are deferred to Appendix A.

In its Lagrangian formulation, pessimistic DPO adds a forward KL term to the DPO objective (§3.3). For the sake of analysis, we assume that the preference annotations are sampled from the reference distribution, $\mu(x) \times \pi_{\mathrm{ref}}(y \mid x) \times \pi_{\mathrm{ref}}(y \mid x)$. Then a finite-sample approximation of the forward KL term is $\hat{\Omega}(\Theta) := \sum_{(y^w, y^\ell) \in \mathcal{D}_{\mathrm{Pref}}} -(\log \pi_\theta(y^\ell) + \log \pi_\theta(y^w))$. By applying this finite-sample approximation, *p-DPO has a finite optimum, unlike DPO*, as shown in Proposition 1. Note that this analysis is limited in two ways: (1) as mentioned, we compute the KL term over the completions in the preference data; (2) we directly optimize the probability ratios $\psi_w = \pi_\theta(y^w)/\pi_{\mathrm{ref}}(y^w)$ and $\psi_\ell = \pi_\theta(y^\ell)/\pi_{\mathrm{ref}}(y^\ell)$, rather than optimizing them jointly through the parameters. For sufficiently expressive $\pi_\theta$, however, this approximation captures the behavior of the two algorithms reasonably well.

**Proposition 3.** *Let $\hat{\mathcal{L}}_{\mathrm{pdpo}}$ represent a finite-sample approximation to $\mathcal{L}_{\mathrm{pdpo}}$ with the empirical forward KL term $\hat{\Omega}(\Theta)$. For a fixed $\hat{\pi}_\theta(y_i^w)$ and $\alpha > 1$, the $\mathrm{argmin}_{\pi_\theta(y^\ell)} \hat{\mathcal{L}}_{\mathrm{pdpo}}$ is* $\min\left(1 - \hat{\pi}_\theta(y_i^w), \hat{\pi}_\theta(y_i^\ell)\right)$, *with* $\log \hat{\pi}_\theta(y_i^\ell) = -\frac{1}{\beta} \log(\alpha - 1) + \log \hat{\pi}_\theta(y_i^w) + \log \frac{\pi_{\mathrm{ref}}(y_i^\ell)}{\pi_{\mathrm{ref}}(y_i^w)}$.

The optimum in Proposition 3 corresponds to $\log \psi_w/\psi_\ell = \beta^{-1} \log(\alpha - 1)$. Recall that IPO seeks to assign a constant value to this ratio by minimizing $(\log \frac{\psi_w}{\psi_\ell} - \tau^{-1})^2$; the (unconstrained) optima are identical for $\tau^{-1} := \beta^{-1} \log(\alpha - 1)$, but the loss surfaces are different (see Appendix B). DPO sets $\pi_\theta(y_i^\ell) \to 0$, as shown in Corollary 1; this is due not only to competition from $\pi_\theta(y_i^w)$ but from DPO penalizing positive probability on $y_i^\ell$. Analysis of the distilled loss gives a similar result:

**Proposition 4.** *For any fixed $\hat{\pi}_\theta(y_i^w)$ and $\beta > 0$, the $\mathrm{argmin}$ of the distilled DPO objective (eq. (7)) is* $\min(1 - \hat{\pi}_\theta(y_i^w), \hat{\pi}_\theta(y_i^\ell)$, *with* $\log \hat{\pi}_\theta(y_i^\ell) = \frac{1}{\beta}(r_t(x, y_i^\ell) - r_t(x, y_i^w)) + \log \hat{\pi}_\theta(y_i^w) + \log \frac{\pi_{\mathrm{ref}}(y_i^\ell)}{\pi_{\mathrm{ref}}(y_i^w)}$.

While the setting is simplistic, the results are comforting: here the additional regularization effects of both distillation and pessimism (in the case of p-DPO) clearly help to avoid degenerate optima.

**Why DPO can drive $\pi(y^w)$ to zero.**    In §2.3 we pointed out a peculiarity of the DPO global optima: in certain cases, it can include policies where $\pi(y^w)$ may be nearly 0 for all $y^w$ in the training set. This undesirable behavior has also been observed in practice [20; 22; 30]. For intuition on why this may happen, consider the simplified case where the policy is a bag-of-words model, $\pi_\theta(y) \propto \exp(c(y) \cdot \theta)$ for $c(y)$ representing a vector of counts in $y$ and $\theta_i$ representing the unnormalized log-probability of token $i$. Then we can formally show that DPO optimization monotonically decreases an upper bound on the probability of the *preferred* completion, $\tilde{\pi}_{\theta^{(t-1)}}(y^w) \geq \tilde{\pi}_{\theta^{(t)}}(y^w) \geq \pi_{\theta^{(t)}}(y^w)$.

**Proposition 5.** *Let $y^w, y^\ell \in \mathcal{V}^n$ be preferred vs. dispreferred outputs of length $n$, with $\pi_{\mathrm{ref}}(y^w), \pi_{\mathrm{ref}}(y^\ell) > 0$ and corresponding count vectors $c(y^w), c(y^\ell)$. Let $\log \pi_\theta(y) = c(y) \cdot \theta - nZ(\theta)$ for $Z(\theta) = \log \sum_i^{\mathcal{V}} e^{\theta_i}$, with upper bound $\log \tilde{\pi}_\theta(y) = c(y) \cdot \theta - n \max_j \theta_j$. Let $\theta^{(t)}$ represent the parameters of $\pi$ after $t$ steps of gradient descent on $\mathcal{L}_{\mathrm{dpo}}(\{y^\ell, y^w, x\})$, with $\theta^{(0)} = 0$. Then $\pi_{\theta^{(t)}}(y^w) \leq \tilde{\pi}_{\theta^{(t)}}(y^w) \leq \tilde{\pi}_{\theta^{(t-1)}}(y^w)$ for all $t$.*

**Where does the probability mass go?**    If $\pi_{\theta^{(t)}}(y^w)$ decreases in $t$, what other strings become more probable? In the following proposition, we show that under the bag-of-words model, DPO optimization moves probability mass away from $y^w$ to sequences that contain only the tokens that maximize the difference between $y^w$ and $y^\ell$. This is a concrete example of the type of undesirable optima described in §2.3, now shown here to be realizable.

---

[4]We also experimented with an ensemble where members are different checkpoints across training of a reward model on the preference data and did not observe any empirical gains from this form of ensemble.

**Proposition 6.** *Let $y^w$ and $y^\ell$ be preferred / dispreferred outputs of length $n$. Let $\Delta = c(y^w) - c(y^\ell)$ be the difference in unigram counts. Let $\hat{y} = [i, i, \ldots, i]$, for $i \in \arg\max \Delta$, with $||c(\hat{y})||_1 = n$. Then $\pi_{\theta(t)}(y^w) - \pi_{\theta(t)}(\hat{y}) = \tau(t)k$ for some $k \leq 0$ and some non-decreasing $\tau : \mathbb{Z}_+ \to \mathbb{R}_+$.*

We have $k = 0$ when $c(y^w) = c(\hat{y})$, and $k \ll 0$ when $||c(y^w)||_2 \ll ||c(\hat{y})||_2 = n$ (dense $c(y^w)$) and $||\Delta||_2 = ||\Delta||_\infty$ (sparse $\Delta$). This implies that when $y^w$ and $y^\ell$ are similar, $\pi_\theta(y^w)$ will degrade more rapidly. Early stopping will therefore tradeoff between reaching the degenerate solution on such cases, and underfitting other cases in which $y^w$ and $y^\ell$ are more distinct.

# 6 Related work

Recent work in offline alignment has focused on DPO [23] as a simpler alternative for aligning language models from preference data. Subsequent work has identified issues with DPO, including weak regularization [3] and a tendency to decrease the probability of winning generations during training [20]. Other methods have explored various avenues for improvement. These include analyzing the impact of noise on DPO alignment [11], proposing to update the reference policy during training [12], and suggesting a variant of IPO with a per-context margin [1]. Additional research has focused on token-level alignment methods [38; 22] and on developing a unified view of various offline alignment methods [31]. This work builds upon several these findings, and provides further analysis, as well as a solution based on pessimism and reward distillation.

While offline alignment methods are popular, recent evidence suggests that online alignment methods such as RLHF [6; 29], may lead to more favorable outcomes [13; 30; 8; 34]. Notably, Zhu et al. [41] proposed iterative data smoothing, which uses a trained model to softly label data during RLHF. Whether online or offline, however, policies are still succeptible to overfitting to certain degenerate phenomena. To this end, reward ensembles have been widely investigated recently as a mechanism for tackling reward hacking in RLHF [9; 7; 39; 25], and in the context of multi-objective optimization [19; 24]. We use an ensemble of rewards to represent the uncertainty with respect to reward models that are suitable given preference data. Moskovitz et al. [19] focus on "composite" rewards, with the goal of achieving high task reward while ensuring that every individual component is above some threshold—also by applying a Lagrangian relaxation. In this work, we also consider multiple reward models, but we only focus on cases where there is no known, obvious reward decomposition.

Finally, the question of using a small amount of offline data to learn high-quality policies, instead of online access to reward feedback, has been widely studied in the offline reinforcement learning (RL) literature. The predominant approach here is to use pessimism, that is, to learn a policy with the highest reward under all plausible environment models consistent with the data, with an extensive theoretical [18; 37; 33] and empirical [16; 5; 36] body of supporting work. The key insight in this literature is that without pessimism, the RL algorithm learns undesirable behaviors which are not explicitly ruled out in the training data, and pessimism provides a robust way of preventing such undesirable extrapolations, while still preserving generalization within the support of the data.

# 7 Conclusion

LM alignment is crucial for deploying safe and helpful assistants, but is difficult due to lack of access to perfect preference oracles. We presented a thorough theoretical analysis of some of the degeneracies that DPO is susceptible to when learning from sampled human preference data. Furthermore, our findings suggest that explicit reward modeling remains a powerful vehicle for introducing regularization into post-training. By distilling the reward assigned by a single, explicit reward model—or a family of explicit reward models—directly into the implicit reward maximized by our policies using offline data, we demonstrated that we can achieve improved robustness to variations in preference dataset quality, while maintaining the simplicity of the DPO framework.

**Limitations.** The empirical results in the paper are based on one dataset and form of distribution shift. For deeper understanding of pessimism and ensembling, additional settings should be explored. The theoretical aspects of the paper are sometimes based on restrictive assumptions and simplifications. Nonetheless, they provide potential explanations for phenomena observed in real-world settings.

**Broader impact.** We introduce new ideas to the active field of research on preference-based post-training, which we hope will help facilitate the alignment of large models, and improve understanding of current approaches—ultimately supporting the development of capable and reliable AI systems.

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

## A Proofs

### A.1 Proof of Proposition 1

*Proof.* Since all the preference pairs $(y, y')$ are mutually disjoint, and $\theta_y$ is specific to each $y$, the DPO objective over $\mathcal{D}_{\mathrm{pref}}$ is convex in $\Delta = \{\Delta_1, \ldots, \Delta_n\}$, where

$$\Delta_i = \beta \log \frac{\pi_\theta(y_i^w)\pi_{\mathrm{ref}}(y_i^\ell)}{\pi_\theta(y_i^\ell)\pi_{\mathrm{ref}}(y_i^w)}. \tag{15}$$

Furthermore, the different $\Delta_i$ are completely independent from each other due to the preference pairs being disjoint, so they can be optimized over separately.

In particular, for every $i$ we have that

$$\lim_{\Delta_i \to \infty} - \log\left(\sigma\left(\Delta_i\right)\right) = 0, \tag{16}$$

which implies that $\Delta^* = \{\infty\}^n$ is the unique global minimizer of the DPO loss over $\mathcal{D}_{\mathrm{pref}}$ in the space of $\Delta$'s, and any $\theta^*$ that is a global minimizer must therefore satisfy

$$\log \frac{\pi_\theta(y_i^w)\pi_{\mathrm{ref}}(y_i^\ell)}{\pi_\theta(y_i^\ell)\pi_{\mathrm{ref}}(y_i^w)} = \infty. \tag{17}$$

$\square$

### A.2 Proof of Corollary 1

*Proof.* Following the same argument of the proof of Proposition 1, we have that all global minimizers $\theta^*$ of the DPO satisfy $\Delta_i^* = \infty$, which in turn implies that

$$\frac{\pi_{\theta^*}(y_i^w)\pi_{\mathrm{ref}}(y_i^\ell)}{\pi_{\theta^*}(y_i^\ell)\pi_{\mathrm{ref}}(y_i^w)} = \infty. \tag{18}$$

Since $\pi_{\mathrm{ref}}(y)$ is assumed to satisfy $0 < \pi_{\mathrm{ref}}(y) < 1$ for all $y$, this implies that all $\theta^*$ satisfy

$$\frac{\pi_{\theta^*}(y_i^w)}{\pi_{\theta^*}(y_i^\ell)} = \infty, \tag{19}$$

which further implies that $\pi_{\theta^*}(y_i^\ell) = 0$ and $\pi_{\theta^*}(y_i^w) > 0$ for all $i \in [n]$, as $\pi_{\theta^*}(y_i^w) \leq 1$ for any $y_i^w$. Aggregating

$$\mathcal{C}(y_\ell) = \{y \colon \exists i \in [n] \text{ s.t } y_i^\ell = y\} \tag{20}$$

then gives that

$$\pi_{\theta^*}(\mathcal{C}(y_\ell)) = \sum_{y \in \mathcal{C}(y_\ell)} \pi_{\theta^*}(y) = 0 \Longrightarrow \pi_{\theta^*}(\mathcal{C}(y_\ell)^c) = 1. \tag{21}$$

$\square$

To prove the converse, let $\pi_{\theta'}$ be a policy that satisfies $\pi_{\theta'}(\mathcal{C}(y^\ell)^c) = 1$, with $\pi_{\theta'}(y_i^w) > 0, \forall i \in [n],$. As $\pi_{\theta'}(y) \geq 0$ for all $y$, this implies that $\pi_{\theta'(y_i^\ell)} = 0 \; \forall i \in [n]$. Then, we have

$$\frac{\pi_{\theta'}(y_i^w)}{\pi_{\theta'}(y_i^\ell)} = \infty, \tag{22}$$

which by Proposition 1 implies that $\pi_{\theta'}$ is a global optimum.

### A.3 Proof of Theorem 1

*Proof.* We know that the optimal policy for the RLHF objective (1) is given by $\pi_{\theta^*}(y|x) \propto \pi_{\mathrm{ref}}(y|x) \exp(r^*(x, y)/\beta)$. Plugging this policy into the distillation objective (7), we see that $\mathcal{L}_{\mathrm{distill}}(r^*, \pi_{\theta^*}, \rho) = 0$ for all $\rho$. In fact, the loss is equal to 0 pointwise, meaning that $\pi_{\theta^*}$ is

543 a global minimizer of the distillation objective (7). Further, let $\pi$ be some other minimizer of
544 $\mathcal{L}_{\text{distill}}(r^*, \cdot, \rho)$. Then $\pi$ also has to attain a loss of $0$ at all $(x, y, y')$ in the support of $\rho$, meaning
545 that $\log \pi(y|x) - \log \pi(y'|x) = \log \pi_{\theta^*}(y|x) - \log \pi_{\theta^*}(y|x)$ for all $(x, y, y')$ in the support of $\rho$.
546 Consequently, the two policies coincide in the support of $\rho$ (due to the normalization constraint, there
547 is no additional offset term allowed as the support of $\rho$ covers all of $\mathcal{Y}$). Finally, noting that the
548 support of the chosen $\rho$ is such that $\pi_{\theta^*}$ puts no mass outside its support due to the KL constraint
549 in (1), we complete the proof. $\qquad\square$

## A.4 Proof of Theorem 2

551 *Proof.* Consider the pessimistic objective:

$$\max_{\pi_\theta} \min_{r_{\text{tgt}} \in \mathcal{S}} \mathbb{E}_{\mu(x)} \Big[ \mathbb{E}_{\pi_\theta(y|x)}[r_{\text{tgt}}(x, y)] - \mathbb{E}_{\pi_{\text{ref}}(y|x)}[r_{\text{tgt}}(x, y)] \Big] - \beta \mathbb{D}_{\text{KL}}(\pi_\theta \| \pi_{\text{ref}}). \tag{23}$$

552 As it is linear in $r_{\text{tgt}}$ and convex in $\pi$, we can switch the order of $\min$ and $\max$:

$$\min_{r_{\text{tgt}} \in \mathcal{S}} \left[ \max_{\pi \in \Pi} \mathbb{E}_{\mu(x)} \Big[ \mathbb{E}_{\pi(y|x)}[r_{\text{tgt}}(x, y)] - \mathbb{E}_{\pi_{\text{ref}}(y|x)}[r_{\text{tgt}}(x, y)] \Big] - \beta \mathbb{D}_{\text{KL}}(\pi \| \pi_{\text{ref}}) \right]. \tag{24}$$

553 Note that every $r_{\text{tgt}} \in \mathcal{S}$ can be written in terms of the KL-constrained policy $\pi_{r_{\text{tgt}}}^*$ it induces, i.e.,

$$r_{\text{tgt}}(x, y) = \beta \log \frac{\pi_{r_{\text{tgt}}}^*(y \mid x)}{\pi_{\text{ref}}(y \mid x)} + \beta \log Z(x, r_{\text{tgt}}), \tag{25}$$

554 where

$$\pi_{r_{\text{tgt}}}^* = \operatorname*{argmax}_{\pi_\theta} \mathbb{E}_{\mu(x)} \mathbb{E}_{\pi_\theta(y|x)}[r_{\text{tgt}}(x, y)] - \beta \mathbb{D}_{\text{KL}}(\pi_\theta \| \pi_{\text{ref}}) \tag{26}$$

555 which has the form

$$\pi_{r_{\text{tgt}}}^*(y \mid x) = \frac{1}{Z(x, r_{\text{tgt}})} \pi_{\text{ref}}(y \mid x) \exp\left( \frac{1}{\beta} r_{\text{tgt}}(x, y) \right) \tag{27}$$

556 where $Z(x, r_{\text{tgt}})$ is the partition function:

$$Z(x, r_{\text{tgt}}) = \sum_{y \in \mathcal{Y}} \pi_{\text{ref}}(y \mid x) \exp\left( \frac{1}{\beta} r_{\text{tgt}}(x, y) \right). \tag{28}$$

557 Substituting $\pi_{r_{\text{tgt}}}^*$ in for $\max_\pi$ and writing $r_{\text{tgt}}$ in terms of $\pi_{r_{\text{tgt}}}^*$, we get the simplified objective

$$
\begin{aligned}
\min_{r_{\text{tgt}} \in \mathcal{S}} & \left[ \max_{\pi \in \Pi} \mathbb{E}_{\mu(x)} \Big[ \mathbb{E}_{\pi(y|x)}[r_{\text{tgt}}(x, y)] - \mathbb{E}_{\pi_{\text{ref}}(y|x)}[r_{\text{tgt}}(x, y)] \Big] - \beta \mathbb{D}_{\text{KL}}(\pi \| \pi_{\text{ref}}) \right] \\
&= \min_{r_{\text{tgt}} \in \mathcal{S}} \left[ \mathbb{E}_{\mu(x)} \left[ \mathbb{E}_{\pi_{r_{\text{tgt}}}^*(y|x)} \left[ \beta \log \frac{\pi_{r_{\text{tgt}}}^*(y \mid x)}{\pi_{\text{ref}}(y \mid x)} + \beta \log Z(x, r_{\text{tgt}}) \right] \right. \right. \\
&\qquad\qquad - \mathbb{E}_{\pi_{\text{ref}}(y|x)} \left[ \beta \log \frac{\pi_{r_{\text{tgt}}}^*(y \mid x)}{\pi_{\text{ref}}(y \mid x)} + \beta \log Z(x, r_{\text{tgt}}) \right] \\
&\qquad\qquad \left. \left. - \beta \mathbb{D}_{\text{KL}}(\pi_{r_{\text{tgt}}}^* \| \pi_{\text{ref}} \mid x) \right] \right] \\
&= \min_{r_{\text{tgt}} \in \mathcal{S}} \beta \left[ \mathbb{E}_{\mu(x)} \Big[ \mathbb{D}_{\text{KL}}(\pi_{r_{\text{tgt}}}^* \| \pi_{\text{ref}} \mid x) + \mathbb{D}_{\text{KL}}(\pi_{\text{ref}} \| \pi_{r_{\text{tgt}}}^* \mid x) - \mathbb{D}_{\text{KL}}(\pi_{r_{\text{tgt}}}^* \| \pi_{\text{ref}} \mid x) \Big] \right] \\
&= \min_{r_{\text{tgt}} \in \mathcal{S}} \beta \mathbb{E}_{\mu(x)} \Big[ \mathbb{D}_{\text{KL}}(\pi_{\text{ref}} \| \pi_{r_{\text{tgt}}}^* \mid x) \Big].
\end{aligned}
\tag{29}
$$

558 $\qquad\square$

### A.5 Proof of Proposition 2

*Proof.* The proof is a standard Lagrangian duality argument, which we reproduce here for completeness. For two functions $f(z)$ and $g(z)$, let us define

$$z^* = \arg\min_z f(z) + \alpha g(z). \tag{30}$$

Let us also consider the constrained problem

$$z' = \arg\min_z f(z) \quad \text{s.t. } g(z) \le g(z^*). \tag{31}$$

Suppose by contradiction that $z^*$ is not a minimizer of (31). Since $z^*$ is feasible for the constraint by construction, we get that $f(z') < f(z^*)$. Consequently, we further have

$$f(z') + \alpha g(z') < f(z^*) + \alpha g(z^*),$$

where the inequality follows from the feasibility of $z'$ in (31). This contradicts the optimality of $z^*$ in (30), meaning that $z^*$ must be a minimizer of (31). Applying this general result with $f = \beta \mathbb{E}_{\mu(x)} \mathbb{D}_{\mathrm{KL}}(\pi_{\mathrm{ref}}(y \mid x) \| \pi_\theta(y \mid x))$, $g = \min_{r_{\mathrm{tgt}}^i \in \mathcal{S}} \mathcal{L}_{\mathrm{distill}}(r_{\mathrm{tgt}}^i, \pi_\theta; \rho)$, and $z = \pi_\theta$ completes the proof, since we recognize the set $\mathcal{S}_\gamma$ in (12) to be equivalent to $\bigcup_{r_{\mathrm{tgt}}^i \in \mathcal{S}} \mathcal{L}_{\mathrm{distill}}(r_{\mathrm{tgt}}^i, \pi_\theta; \rho) \le \lambda$.

$\square$

### A.6 Proof of Proposition 3

*Proof.* We differentiate $\mathcal{L}_{\mathrm{pdpo}}$ with respect to $\psi_\ell = \pi_\theta(y^\ell)/\pi_{\mathrm{ref}}(y^\ell)$ with $i$ implicit, obtaining,

$$\frac{\partial \mathcal{L}_{\mathrm{pdpo}}}{\partial \psi_\ell} = \beta \frac{\psi_\ell^\beta}{\psi_w^\beta + \psi_\ell^\beta} \psi_\ell^{-1} - \frac{\beta}{\alpha} \psi_\ell^{-1} = \beta \psi_\ell^{-1} \left( \frac{\psi_\ell^\beta}{\psi_w^\beta + \psi_\ell^\beta} - \alpha^{-1} \right) \tag{32}$$

which is zero when,

$$\alpha \psi_\ell^\beta = \psi_w^\beta + \psi_\ell^\beta \tag{33}$$

$$\psi_\ell = \left( \frac{1}{\alpha - 1} \right)^{1/\beta} \psi_w \tag{34}$$

$$\log \psi_\ell = -\frac{1}{\beta} \log(\alpha - 1) + \log \psi_w \tag{35}$$

$$\log \pi_{\hat{\theta}}(y^\ell) = \log \pi_{\mathrm{ref}}(y^\ell) - \frac{1}{\beta} \log(\alpha - 1) + \log \pi_\theta(y^w) - \log \pi_{\mathrm{ref}}(y^w). \tag{36}$$

By the second-order condition, the critical point is a minimum. The objective $\mathcal{L}_{\mathrm{pdpo}}$ is the sum of two components: the negative log sigmoid term for $\mathcal{L}_i$ and the negative log probability for $\hat{\Omega}$. Because each component is a convex function of $\psi_i$, so is $\mathcal{L}_{\mathrm{pdpo}}$. As a result, the local minimum $\log \hat{\pi}_\theta(y^\ell)$ is also a global minimum. $\square$

### A.7 Proof of Proposition 4

*Proof.* This follows directly from differentiating eq. (7) with respect to $\pi_\theta(y_2)$. $\square$

### A.8 Proof of Proposition 5

*Proof.* Let $\Delta = [c(y^w) - c(y^\ell)]$ and $\rho = \pi_{\mathrm{ref}}(y^w)/\pi_{\mathrm{ref}}(y^\ell)$. The theorem assumes $|y^w| = |y^\ell|$. Then $\mathcal{L}_{\mathrm{dpo}} = -\log \sigma \left( \beta(\Delta \cdot \theta) + \beta \log \rho \right)$. The derivative with respect to $\theta$ is,

$$\frac{\partial \mathcal{L}_\beta(\theta)}{\partial \theta} = -(1 - \sigma(\beta(\Delta \cdot \theta) + \beta \log \rho))\beta\Delta = -\Pr(y^\ell \succ y^w; \theta)\beta\Delta \prec 0. \tag{37}$$

Let $\delta_t = \beta \Pr(y^\ell \succ y^w; \theta^{(t)})$. Then,

$$\tilde{\pi}_{\theta^{(t)}} = \theta^{(t)} \cdot c(y^w) - n \max_j \theta_j^{(t)} \tag{38}$$

$$= (\theta^{(t-1)} + \delta_t \Delta) \cdot c(y^w) - n \max_j (\theta_j^{(t-1)} + \delta_t \Delta_j) \tag{39}$$

$$= \theta^{(t-1)} \cdot c(y^w) - n \max_j \theta_j^{(t-1)} + \delta_t \Delta \cdot c(y^w) - n\delta_t \max_j \Delta_j \tag{40}$$

$$= \tilde{\pi}_{\theta^{(t-1)}} + \delta_t \left( \Delta \cdot c(y^w) - n \max_j \Delta_j \right) \tag{41}$$

$$= \tilde{\pi}_{\theta^{(t-1)}} + \delta_t \sum_j^{\mathcal{V}} c_j(y^w)(\Delta_j - \max_{j'} \Delta_{j'}) \leq \tilde{\pi}_{\theta^{(t-1)}}. \tag{42}$$

We obtain $\max_j \left( \theta_j^{(t-1)} + \delta_t \Delta_j \right) = \max_j \theta_j^{(t-1)} + \max_j \delta_t \Delta_j$ from the fact that $\theta^{(0)} = 0$ and therefore $j \in \arg\max \Delta$ implies $j \in \arg\max \theta^{(t')}$ for all $t' > 0$. The second-to-last step uses $n = \sum_j^{\mathcal{V}} c_j(y^w)$ and the final step uses $\Delta_j \leq \max_{j'} \Delta_{j'}$. Finally, we have $\pi_{\theta^{(t)}}(y) \leq \tilde{\pi}_{\theta^{(t)}}(y^w)$ because $Z(\theta) = \log \sum_j \exp \theta_j \geq \log \max_j \exp \theta_j = \max_j \theta_j$. $\qquad\square$

### A.9 Proof of Proposition 6

*Proof.* Applying gradient descent with learning rate $\eta$ to the gradient from Equation (37), at each step $t$ the parameters are,

$$\theta^{(t)} = \theta^{(t-1)} + \eta\beta \Pr(y^\ell \succ y^w; \theta^{(t-1)})\Delta = \left( \sum_{t'=1}^{t} \eta\beta \Pr(y^\ell \succ y^w; \theta^{(t')}) \right) \Delta = \tau(t)\Delta. \tag{43}$$

Plugging these parameters into the likelihoods,

$$\ell_{\theta^{(t)}}(c(y^w)) - \ell_{\theta^{(t)}}(\hat{y}) = c(y^w) \cdot \theta^{(t)} - nZ(\theta^{(t)}) - c(\hat{y}) \cdot \theta^{(t)} + nZ(\theta^{(t)}) \tag{44}$$

$$= (c(y^w) - c(\hat{y})) \cdot \theta^{(t)} = (c(y^w) - c(\hat{y})) \cdot (\tau(t)\Delta) \tag{45}$$

$$= \tau(t)(c(y^w) \cdot \Delta - n \max \Delta) = \tau(t)k, \tag{46}$$

with $k \leq 0$ by $c(y^w) \cdot \Delta \leq ||c(y^w)||_1 \times ||\Delta||_\infty = n \max \Delta$. $\qquad\square$

## B  Transitive closure

Both p-DPO and IPO target a constant ratio for $\log \psi_w/\psi_l$. However, the loss surfaces are different. To see this, we consider a simplified setting with three possible outputs, $y_1, y_2, y_3$. We observe either $\mathcal{D} = \{(y_1 \prec y_2), (y_2 \prec y_3)\}$ or $\overline{\mathcal{D}} = \mathcal{D} \cup \{(y_1 \prec y_3)\}$. If we treat this problem as a multi-arm bandit, the goal is to assign a weight to each arm, which we denote $\psi_i = \log \pi_\theta(y_i|x) + Z_x$, with $Z_x$ an underdetermined log-partition function.

**Proposition 7.** *Let $\mathcal{D} = \{(i, i+1) : i \in 1, 2, \ldots, n\}$ for $n > 2$. Let $\overline{\mathcal{D}}$ be the dataset arising from the transitive closure of $\mathcal{D}$. Assume $\pi_{\text{ref}}$ is indifferent to all $(y_i, y_j)$. Let $\psi_\infty^{(\mathcal{D})} = \max_i \psi_i^{(\mathcal{D})} - \min_i \psi_i^{(\mathcal{D})}$. Then $\psi_\infty^{(\mathcal{D})} = (n-1)\tau^{-1} > \psi_\infty^{(\overline{\mathcal{D}})} = 2\frac{n-1}{n}\tau^{-1}$.*

*Proof.* For $\mathcal{D}$, the IPO objective can be minimized at zero, so that $\psi_\infty^{(\mathcal{D})} = (n-1)\tau^{-1}$. For $\overline{\mathcal{D}}$, each adjacent pair of completions is separated by $\gamma$, and the objective is $\sum_{i=1}^{n-1}(n-i)(i\gamma - \tau^{-1})^2$. The minimum is $\gamma = \frac{n(n+1)(n-1)/6}{n^2(n+1)(n-1)/12}\tau^{-1} = \frac{2}{n}\tau^{-1}$, so that $\psi_\infty^{(\overline{\mathcal{D}})} = (n-1)\gamma = 2\frac{n-1}{n}\tau^{-1} < (n-1)\tau^{-1} = \psi_\infty^{(\mathcal{D})}$ for $n > 2$. $\qquad\square$

Intuitively, the observation of $(y_1 \prec y_3)$ should increase confidence that $y_3$ is superior to $y_1$, but in IPO it has the opposite effect, drawing their scores closer together. While pessimistic DPO also has a target ratio between each preference pair, its loss surface is different: in particular, it does not

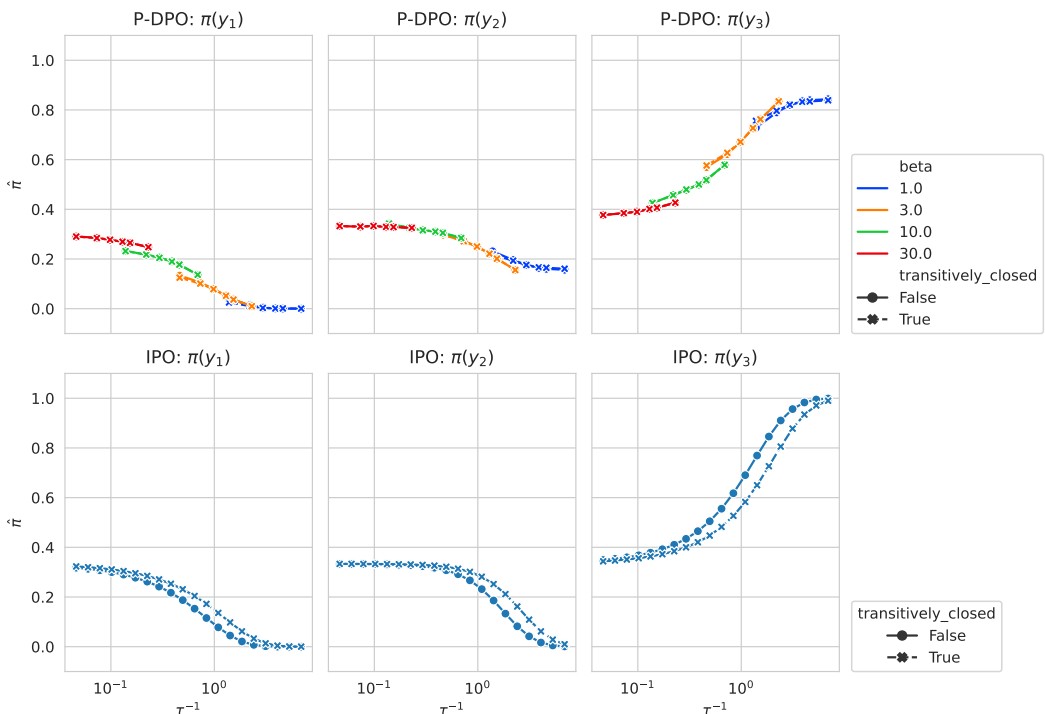

Figure B.1: **Effect of transitive closure on p-DPO and IPO solutions to preference learning in a multi-arm bandit**. Each column shows the learned policy probability for a given arm, based on the preferences $y_1 \prec y_2 \prec y_3$. The top row shows that in p-DPO, the probabilities are not materially affected by the transitive closure $y_1 \prec y_3$. The bottom row shows that in IPO, transitive closure causes the probabilities to be compressed. In each subfigure, we sweep a range of effective values of $\tau^{-1}$, shown on the x-axis.

increase quadratically as we move away from the target. We find empirically that pessimistic DPO is robust to the transitive closure of preference annotations in the multi-arm bandit setting, as shown in Figure B.1. As discussed above, DPO will set $\psi_1 \to -\infty$ because $y_1$ is never preferred.

In our empirical experiments we solve the p-DPO and IPO objectives for both $\mathcal{D} = \{(y_1, y_2), (y_2, y_3)\}$ and $\overline{\mathcal{D}} = \mathcal{D} \cup \{(y_1, y_3)\}$, solving with respect to $\{\pi_\theta(y_i)\}$. IPO is solved analytically as a quadratic program; for pessimistic DPO we used projected gradient descent. We consider $\beta \in (1, 3, 10, 30)$ and $\alpha \in (5, 10, 20, 50, 100, 1000)$. As shown in Figure B.1, there are significant differences in the IPO solutions with and without transitive closure, while for p-DPO these differences are imperceptible.

## C   Distribution over reward models for e-DPO

Figure C.1 investigates the reason for the success of e-DPO, especially when $\rho < .5$. For every length bias, we show across all training examples the fraction of cases where a certain reward model, $r_{\rho,b}$, best matched the implicit reward of the final e-DPO policy. The policy matches different reward models in different examples. Moreover, there is inverse correlation between the data bias for policy training ($\rho$) and the data bias for training the reward models ($b$). This suggests that the ensemble in e-DPO helps as the policy is distilling from reward models that do not share the data bias of the policy training set.

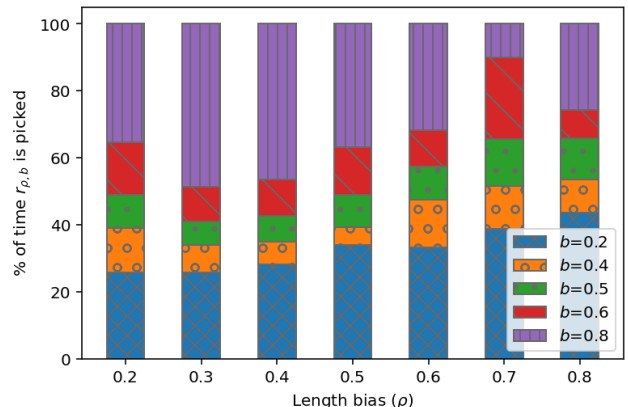

Figure C.1: We show for every length bias, $\rho$, the distribution over reward models that best match the final policy trained by e-DPO across all training examples. We observe that the e-DPO policy matches different reward models across examples. Moreover, when the policy is trained with data biased towards preferring short responses, the reward model that was trained on longer responses is often preferred and vice versa.

## D Hyperparameters

Validation set performance across the range of hyperparameter settings is shown in Figure D.1. In pilot studies we found that these results were relatively robust to variation in the random seed, but did not conduct extensive investigation of this effect across all methods and hyperparameters due to cost.

## E Compute resources

We train policies on 32 TPU v3 chips and reward models on 16 TPU v3 chips. We obtain roughly 0.1 steps per second when training, for both the policy and reward models.

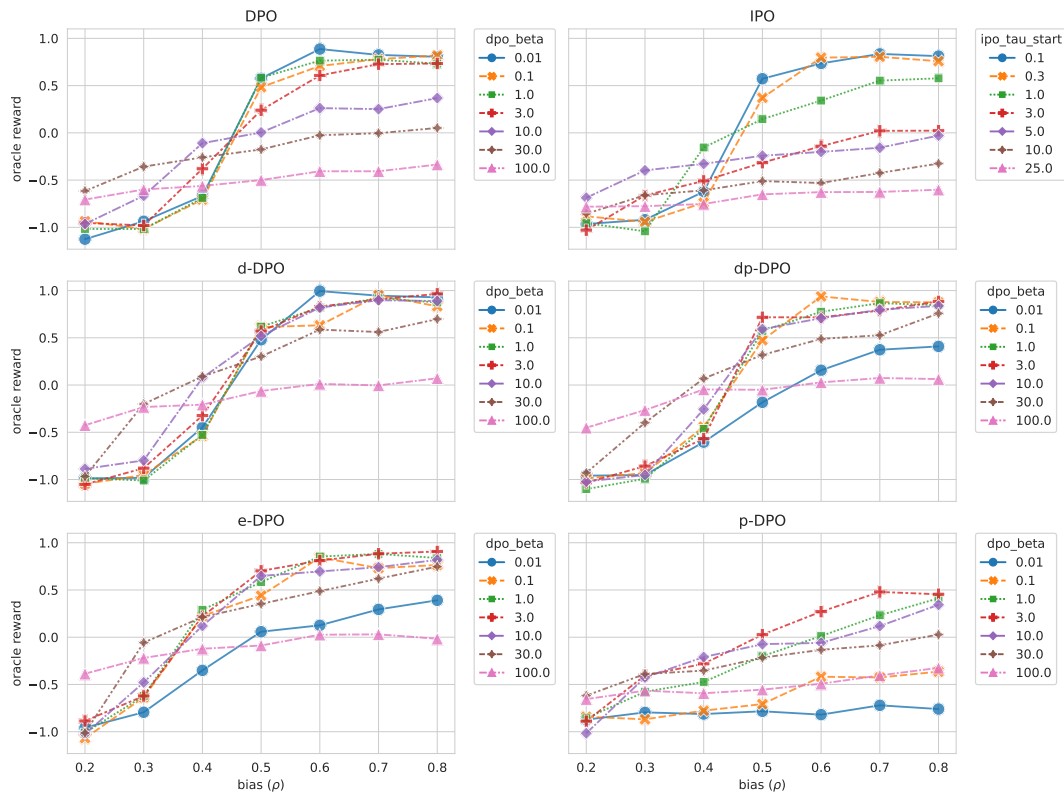

Figure D.1: **Validation set results** across hyperparameters for each method. For all methods, different values of $\rho$ induce different optimal hyperparameters $\beta$ and $\tau^{-1}$.

