# OpenReview forum: "Robust Preference Optimization through Reward Model Distillation"
_NeurIPS.cc/2024/Conference — Submitted to NeurIPS 2024_

### Official Review · Reviewer_wkMh · 2024-07-11

**Soundness:** 3
**Presentation:** 3
**Contribution:** 2
**Rating:** 6
**Confidence:** 4

**Summary:**

Based on the analysis of the shortcomings of DPO (Section 2.3), the authors proposed a simple reward distillation approach (Section 3.1) to align language models and a pessimistic variant (Section 3.2). These approaches outperform the vanilla DPO.

**Strengths:**

- The analysis in Section 2.3 extends the result in the IPO paper [23].
- The reward distillation approach is simple and natural, giving good empirical results.

**Weaknesses:**

I think that this paper is technically well done. However, it has a limited application scope.

The authors aim to address the shortcomings of offline alignment methods like DPO, while maintaining an offline approach. However, online alignment methods such as PPO or Reinforce-style algorithms naturally alleviate these offline learning issues. This raises a question: Given the authors' findings on the limitations of offline alignment, why not adopt an online alignment method like PPO directly?

Admittedly, the distilled DPO method and its pessimistic variants are less complex than PPO. However, these proposed offline variants introduce additional complexities to vanilla DPO by requiring separate reward models and separate training phases. This makes them more akin to online methods. Furthermore, the experiment section lacks a comparison against online method baselines, leaving it unclear if the distilled-DPO variants offer a better performance--compute tradeoff compared to vanilla online methods like PPO.

To be clear, the paper offers intriguing insights. However, these techniques seem niche. They benefit those who prefer robust generalization in preference optimization without using online data, an assumption not very well communicated in the paper.

In my opinion, there are ways to increase the method's applicability. While the proposed distillation approaches are positioned as offline methods, to my understanding, they are also applicable to an online regime. For example, in Equation (7), instead of sampling $(x, y_1, y_2)$ from an offline dataset, as the authors currently do, we can sample $x$ from an offline dataset and then sample $(y_1, y_2)$ from the model in training. The learned models, in this way, are directly comparable to those learned via online alignment methods like PPO, but in a simpler approach than these online baselines. If the authors could demonstrate that these models outperform PPO or require less complexity in training, they could expand the method's application scope and impact.

**Questions:**

As mentioned in the "Weaknesses" section above, does it make sense to apply reward distillation to online policy optimization? Does this compare favorably with online policy optimization baselines, like PPO?

**Limitations:**

See "Weaknesses" section above.

---

> ### Author Rebuttal · Authors · 2024-08-07
>
> We thank the reviewer for their careful review and thoughtful comments. We were pleased to see that the reviewer appreciated the technical contribution of this work, and that the reviewer believed that the paper offered intriguing insights. We also appreciate the concerns and questions raised in the review, and are eager to clarify a number of key points:
>
> 1. Reward hacking and optimization issues are present in both online and offline settings. That said, online settings do lend themselves more naturally to modifications to the reward used. This work brings some of this ability back to the offline setting: specifically through the ability to distill from any specially designed reward, as well as concepts of reward pessimism.
> 2. We believe that the insights and solutions offered here are valuable to the community at large. In particular, offline DPO was reported as the key alignment strategy behind Meta’s Llama 3, mainly due to its simplicity. We retain the same simple framework with improved robustness—as well as modest gains on standard settings as our additional experiment in the supplement shows.
>
> We now provide answers to specific questions and remarks (quoted) below.
>
> > The authors aim to address the shortcomings of offline alignment methods like DPO, while maintaining an offline approach. However, online alignment methods such as PPO or Reinforce-style algorithms naturally alleviate these offline learning issues. This raises a question: Given the authors' findings on the limitations of offline alignment, why not adopt an online alignment method like PPO directly?
>
> While online alignment doesn't suffer from the same shortcomings of offline alignment, it is still susceptible to __reward hacking__ and can be difficult to optimize, as has been shown in the past (e.g., Tang et. al., 2024, Eisenstein et. al. 2023, Coste et. al., 2023). Thus, online learning is not a panacea. Moreover, offline methods are both popular (Meta’s recent Llama 3 uses offline DPO for alignment) and have been shown to be effective (e.g., Tajwar et al., 2024). Thus, we argue that there is broad interest in the community in investigating offline methods on their own due to their good performance and simplicity, regardless of the trade-offs compared to methods like PPO.
>
> > The proposed offline variants introduce additional complexities…This makes them more akin to online methods.
>
> The additional overhead of reward distillation is small, and rewards for the training data can be computed offline (once) ahead of time. This allows training to be nearly identical to the standard variants, modulo the structure of the data that is fed in. This is true regardless of the size of the policy that is trained, or its hyperparameters: the reward data is generated once, for a finite (and completely parallelizable) dataset. We therefore respectfully, but strongly, disagree that the proposed method is more akin to online methods, where the reward models score generations sampled online from an ever-shifting distribution.
>
> > The techniques seem niche.
>
> Not only has DPO become popular in academic settings for which slower algorithms like PPO are expensive, but it has also found quite a bit of traction in industry settings. As mentioned, Llama 3 completely relies on multiple iterations of offline DPO—primarily from the stated standpoint that it reduces engineering complexity and provides flexibility for data mixture curation. Our work (a) offers a deeper understanding of DPO than the current literature provides, and (b) a practical algorithm for increasing robustness while preserving the attributes that make it so convenient to train.
>
> > They benefit those who prefer robust generalization.
>
> We focused on a robustness setting in the paper as we view this (and reward hacking more generally) to be a significant (if not the most significant) issue plaguing offline methods like DPO, as has also now been reported by multiple papers in the recent literature (see, e.g., Rafailov et. al, 2024). Pessimism and distillation are principled, but still simple, ways to combat this.
>
> Nevertheless, we have run an additional experiment to test the effectiveness of our method on the __standard, unbiased setting of the Anthropic helpfulness dataset__ (Bai et. al., 2022), see Table 1 of the rebuttal supplement. We use a Gemini Ultra model for evaluating win-rates of the policies over both the SFT starting point and the best DPO baseline. We find that even in this unbiased setting our distillation objectives can provide modest gains in win-rates. Concretely, e-DPO's win rate against the SFT policy is 65.8, while DPO's win rate is 64.2. Moreover, comparing e-DPO and DPO directly, e-DPO wins in 49.7% of the cases, while DPO wins in 46.9% of the cases (the rest are ties). We also note that we also see similar, moderate yet systematic, gains compared to DPO on the unbiased setting (i.e., $\rho = 0.5$) of TL;DR in Figure 2 of the main paper.
>
> On a higher level, a key message of this work is that the community has invested a significant amount of effort over the past years in designing, training, and fixing reward models. Don’t throw them away just yet! They can still be quite useful for offline DPO. We will make this point clearer in the paper, in addition to including the extra results.
>
> Please let us know if this has addressed your concerns. We look forward to engaging in the response period.
>
> [1] Llama team. The Llama 3 Herd of Models. 2024.
>
> [2] Bai et. al. Training a Helpful and Harmless Assistant with Reinforcement Learning from Human Feedback. 2022.
>
> [3] Tang et. al. Understanding the performance gap between online and offline alignment algorithms. 2024.
>
> [4] Tajwar et. al. Preference Fine-Tuning of LLMs Should Leverage Suboptimal, On-Policy Data. 2024.
>
> [5] Coste et. al. Reward model ensembles help mitigate overoptimization. 2023.
>
> [6] Eisenstein et. al. Helping or Herding? Reward Model Ensembles Mitigate but do not Eliminate Reward Hacking. 2023.

---

> > ### Author Response · Authors · 2024-08-09
> >
> > Thanks again for your hard work in reviewing our paper! We hope that you've had a chance to read our reply to your comments. If they've addressed your concerns, then we hope that the reviewer can consider increasing the score. If not, please let us know if you have any further questions or concerns. We are committed to ensuring that our contributions are clearly communicated.

---

> ### Comment · Reviewer_wkMh · 2024-08-12
> **Response**
>
> I appreciate the authors for their response. I especially appreciate the authors' discussion regarding their method's computational complexity:
>
> > The additional overhead of reward distillation is small, and rewards for the training data can be computed offline (once) ahead of time. This allows training to be nearly identical to the standard variants, modulo the structure of the data that is fed in. This is true regardless of the size of the policy that is trained, or its hyperparameters: the reward data is generated once, for a finite (and completely parallelizable) dataset. We therefore respectfully, but strongly, disagree that the proposed method is more akin to online methods, where the reward models score generations sampled online from an ever-shifting distribution.
>
> This corrects my misunderstanding, leading me to adjust my score. I suggest the authors include a similar discussion like this in the paper, emphasizing the overhead is minimal.
>
> The paper currently lacks a comparison between offline methods (like DPO and the authors' approach) and online alignment variants (such as PPO), which I think is a limitation. I understand that incorporating such a comparison within the short rebuttal period might be unrealistic, but I think including it could significantly enhance the paper's impact. Indeed, the core argument of the paper is to address the idiosyncrasies of implicit reward behavior. Since online methods like PPO don't involve implicit rewards, and as the authors noted, "explicit reward models [in offline methods] can easily be regularized and understood," a comparison would seem beneficial. As a reader, I anticipate this comparison. An imperfect analogy might be a paper proposing a vision transformer model robust to shift-translation but neglecting to compare it with a conventional CNN, which is likely to possess this property by design.

---

> > ### Author Response · Authors · 2024-08-13
> >
> > We thank the reviewer for thoroughly reading our author response, and for adjusting their score. We also appreciate and will consider the additional suggestions for improving its contribution. We will certainly incorporate a clear discussion of the computational benefits of our approach in a revision to the main paper.

---

### Official Review · Reviewer_y8pX · 2024-07-11

**Soundness:** 3
**Presentation:** 3
**Contribution:** 3
**Rating:** 6
**Confidence:** 3

**Summary:**

This paper addresses the limitations of DPO in LM alignment by proposing a reward model distillation approach. DPO, while efficient, often leads to overconfident and degenerate policies due to limited preference annotations. The authors introduce a method that trains LMs to match the distribution from a reward model, improving robustness and performance, particularly in biased preference datasets. Their approach also includes a pessimistic extension to handle uncertainty in reward models.

**Strengths:**

This paper makes a considerable theoretical contribution by addressing the critical issue of degeneration in DPO, a problem of significant concern in the community. The authors present a well-structured and clearly written analysis that helps in understanding the overfitting commonly observed with the DPO. The proposed method of reward model distillation is both theoretically sound and intuitive, offering a robust solution that potentially improves upon traditional DPO methods. I appreciate this approach and its theoretical support.

**Weaknesses:**

The main concern about this paper is the evaluation. As discussed in Section 2.3, DPO can shift nearly all probability mass to responses that never appear in the training set, also called OOD extrapolation in other papers. This issue arises when there are few annotations per instance (x, y1, y2). Therefore, it is expected that the authors should demonstrate how their proposed approach mitigates this problem, maybe by presenting the log-probabilities of the winning and losing responses. Specifically, the log-probability margin between winning and losing responses should not keep increasing (I assume this is true?).

Additionally, I can see the paper presents results showing robustness against distribution shifts in preference data. But what is the factor that makes the proposed algorithm learn the policy whose underlying preference distribution closer to the true preference distribution in biased setting? I would be willing to increase my score if these concerns are solved.

**Questions:**

Refer to weaknesses.

**Limitations:**

I have no concerns about the limitations.

---

> ### Author Rebuttal · Authors · 2024-08-07
>
> We thank the reviewer for their careful review and thoughtful comments. We were happy to see that the reviewer viewed our work as a considerable theoretical contribution towards understanding, and mitigating, degeneration in DPO, and that the reviewer thought that our analysis was theoretically sound, well-structured, and clearly written. With respect to the reviewer’s concerns, we provide answers to specific questions and remarks (quoted) below.
>
> > On evaluating shifting probability mass.
>
> In the supplement we test the effect of pessimism in both DPO (DPO vs p-DPO) and distilled DPO (d-DPO vs dp-DPO). See Figure 3. A subtle point arises from the distinction between regularizing to the reference policy output distribution versus regularizing to the preference data distribution, which are only (asymptotically) identical if preferences are annotated on a sample from the reference policy (in our initial experiments we had found slightly better results overall when regularizing to the reference policy output distribution). In the supplement we report results with respect to both distributions, showing that pessimism (a) mitigates the decrease in probability of preferred and dispreferred preference annotations, despite this data not being used in the regularizers (left-most subplots), and (b) mitigates the increase in KL divergence with respect to the reference distribution (third subplot from left), as expected due to the additional regularization term. As we argue in the paper (see also Azar et al 2023), the $\beta$ hyperparameter of DPO does not effectively regularize this KL distribution because the implicit DPO reward model assigns infinite-magnitude rewards.
>
> > On the factors that make the proposed algorithm learn better policies.
>
> The main factors lie in regularization of the policy. First, distillation objectives avoid some of the degeneracies that are theoretically characterized in the paper, simply by virtue of using real-valued reward model signals that are naturally bounded. Second, in Figure 2 of the supplement we have plotted an analysis of the reward models that are picked by the policy (i.e., have the lowest loss per Eq. 10) during training of e-dpo, which uses an ensemble of reward models . This analysis shows that the selected reward models are different for different examples. There is also some tendency for the opposing reward models that are trained at 0.8 and 0.2 length biases to be used more often during training at all length biases. Moreover, the 0.8 reward model is picked more often for length bias 0.2/0.3/0.4, while the 0.2 reward model is picked more often for length bias 0.7/0.8. This suggests (though does not prove) the presence of some positive regularization effect.
>
> Please let us know if this has addressed your concerns. We look forward to engaging in the response period.
>
> [1] Azar et. al. A General Theoretical Paradigm to Understand Learning from Human Preferences. 2023.

---

> > ### Author Response · Authors · 2024-08-09
> >
> > Thanks again for your hard work in reviewing our paper! We hope that you've had a chance to read our reply to your comments. If they've addressed your concerns, then we hope that the reviewer can consider increasing the score. If not, please let us know if you have any further questions or concerns. We are committed to ensuring that our contributions are clearly communicated.

---

> > ### Comment · Reviewer_y8pX · 2024-08-11
> >
> > Thank you to the authors for their thorough rebuttal. My major concerns have been addressed, and with the inclusion of these results in the paper, I believe this paper offers valuable insights to our community. I have accordingly raised my score.

---

> > > ### Author Response · Authors · 2024-08-14
> > >
> > > A quick thank you to the reviewer for reading our author response, and for adjusting their score! We are happy to see that the reviewer believes that our work offers valuable insights to the community. We will be sure to include the new results presented here in the paper.

---

### Official Review · Reviewer_yoUT · 2024-07-17

**Soundness:** 3
**Presentation:** 3
**Contribution:** 3
**Rating:** 6
**Confidence:** 4

**Summary:**

The authors discuss and give formal results on the limitations of DPO that have been observed in practice, and investigate reward model objectives for 1) distilling reward differences into the generator (eq. 7), and 2) pessimistic "minimax" distillation over a family of reward models, to mitigate these limitations. The theoretical equivalance of the "forward" and "reverse" pessimistic formulations (eq. 8 and 9) of the standard RLHF objective (eq. 1) is shown, and an approximate pessimistic objective (eq. 10) with a minimum over distillation losses for the reward model family considered in a Langrangian term. Results on TL;DR preference data that is biased to varying degrees to prefer long responses (Figure 2) shows good results for distilled models generally, and that pessimistic optimization over a family of reward models  optimized for varying response lengths improves results in irregular settings (i.e. when shorter responses are preferred).

**Strengths:**

- A well written, insightful paper, with well formulated, novel objectives for preference optimization.

**Weaknesses:**

- The results, as the authors acknowledge, are currently quite limited. These methods should really be tested on at least one additional preference task, and for results utilizing multiple models, pessimism should be compared with other basic ensembling strategies.
- The gamma parameter is annealed during training, suggesting that the setting is important and perhaps sensitive. Some ablations and discussion around this seems prudent.

**Questions:**

See above sections.

**Limitations:**

Yes.

---

> ### Author Rebuttal · Authors · 2024-08-07
>
> We thank the reviewer for their careful review and thoughtful comments. We were delighted to see that the reviewer finds our work to be well-written, insightful, well-formulated, and novel. Per the reviewer’s suggestions, we have added new experimental results in our supplemental section for this rebuttal period.
>
> Specifically, we ran an additional experiment to test the effectiveness of our method on another preference task — this time in a standard, unbiased setting on the popular Anthropic helpfulness dataset (Bai et. al., 2022), where we use a Gemini Ultra model for evaluating win-rates of the policies over both the SFT starting point and the best DPO baseline. See Table 1 in the supplement. We find that even in this unbiased setting our distillation objectives can still provide modest gains. Concretely, e-DPO's win rate against the SFT policy is 65.8, while DPO's win rate is 64.2. Moreover, comparing e-DPO and DPO directly, e-DPO wins in 49.7% of the cases, while DPO wins in 46.9% of the cases (the rest are ties). Together with our theoretical analysis and strong robustness results, we believe that this makes reward distillation a simple but compelling algorithmic modification to offline training.
>
> In terms of additional ablations, we also tested the effect of the gamma annealing schedule. See Figure 1 of the supplement. We find that this parameter need not be tuned (and we did not extensively tune it in our experiments). As shown in the supplement, a constant gamma of 1e-3 vs. annealing from 1e-4 to 1e-2 leads to very similar results (in fact, even slightly better).
>
> Please let us know if this has addressed your concerns. We look forward to engaging in the response period.
>
> [1] Bai et. al. Training a Helpful and Harmless Assistant with Reinforcement Learning from Human Feedback. 2022.

---

> > ### Author Response · Authors · 2024-08-09
> >
> > Thanks again for your hard work in reviewing our paper! We hope that you've had a chance to read our reply to your comments. If they've addressed your concerns, then we hope that the reviewer can consider increasing the score. If not, please let us know if you have any further questions or concerns. We are committed to ensuring that our contributions are clearly communicated.

---

### Author Rebuttal · Authors · 2024-08-07

Thank you to all the reviewers for taking the time to read and comment on our work. We were delighted to see that overall the reviewers found our work to be well-written, insightful, and a considerable technical contribution. We were also pleased to receive several good questions and suggestions: we have taken them seriously. We have included new experimental results in our supplemental section for this rebuttal period, in addition to our individual responses to specific comments raised by reviewers.

Please let us know if any questions or concerns remain. We look forward to additional discussion.

---

### Decision · Program_Chairs · 2024-09-25

**Decision:**

Reject

**Comment:**

This paper addresses the limitations of DPO in value alignment by proposing a reward model distillation approach. The authors present a method that trains LMs to match the distribution from a reward model to improve robustness and performance, especially in biased preference datasets. Their approach also includes a pessimistic extension to handle uncertainty in reward models. The experiments additionally demonstrate the superiority of the method presented in this paper.

However, some concerns need to be addressed in this paper. All of the reviewers y8px and wkMh suggested that this paper currently lacks a comparison between offline methods and online methods. We recommend the authors compare the methods in this paper with more online methods, such as PPO, Iterative DPO [1,2], and other ensemble strategies. If the authors could demonstrate that these models outperform these methods or require less complexity in training to achieve comparable performance with these methods, they could expand the method's application scope and impact. Moreover, the methods should really be tested on various datasets with multiple popular models, such as Llama-3, Gemma-2, and so on. With these concerns, I believe the current manuscript is not ready for publication and I recommend rejection.

[1] Xiong, Wei, et al. "Iterative preference learning from human feedback: Bridging theory and practice for rlhf under kl-constraint."

[2] Yuan, Weizhe, et al. "Self-rewarding language models."